# Bilateral Symmetry Strengthens the Perceptual Salience of Figure against Ground

**Birgitta Dresp-Langley** 

Centre National de la Recherche Scientifique, ICube Lab UMR 7357 CNRS and Strasbourg University, 67000 Strasbourg, France; birgitta.dresp@icube.unistra.fr or birgitta.dresp@unistra.fr

**Abstract:** Although symmetry has been discussed in terms of a major law of perceptual organization since the early conceptual efforts of the Gestalt school (Wertheimer, Metzger, Koffka and others), the first quantitative measurements testing for effects of symmetry on processes of Gestalt formation have seen the day only recently. In this study, a psychophysical rating study and a "foreground"-"background" choice response time experiment were run with human observers to test for effects of bilateral symmetry on the perceived strength of figure-ground in triangular Kanizsa configurations. Displays with and without bilateral symmetry, identical physically-specified-to-total contour ratio, and constant local contrast intensity within and across conditions, but variable local contrast polarity and variable orientation in the plane, were presented in a random order to human observers. Configurations with bilateral symmetry produced significantly stronger figure-ground percepts reflected by greater subjective magnitudes and consistently higher percentages of "foreground" judgments accompanied by significantly shorter response times. These effects of symmetry depend neither on the orientation of the axis of symmetry, nor on the contrast polarity of the physical inducers. It is concluded that bilateral symmetry, irrespective of orientation, significantly contributes to the, largely sign-invariant, visual mechanisms of figure-ground segregation that determine the salience of figure-ground in perceptually ambiguous configurations.

**Keywords:** bilateral symmetry; physically-specified-to-total contour ratio; Kanizsa's triangle; figure-ground segregation; law of good Gestalt; subjective magnitude estimation; choice response time

## 1. Introduction

The Gestalt psychologists Max Wertheimer and Wolfgang Metzger [1,2] formulated and discussed "laws of perception" to predict how perceptual grouping operates under specific conditions of visual configuration. Their important work was translated into the English language in 2012 and 2009 respectively by Lothar Spillmann and colleagues [1,2], making this important early conceptual work available to a broader audience. In physical science, a law is a prediction that can be proven true and, ideally, the limits of which can also be clearly determined. In perceptual science, the Gestalt laws are used to express principles or conditions of visual configuration to explain why we see the world as we do. It is argued that specific principles of, or conditions for, "good Gestalt" need to be fulfilled to enable what is called perceptual grouping, i.e., a perceptual solution that yields the most plausible interpretation of a given physical configuration. Since all physical stimuli are by nature ambiguous to our perception, they need to be interpreted by the brain to produce coherent and unambiguous percepts that allow us to act on the physical world effectively. The "Law of Symmetry" is a major Gestalt law. It predicts that visual elements that are symmetrical would be more likely to form a group, i.e., to be perceived as a "good Gestalt", in comparison with asymmetrical ones. Visual symmetry has, indeed, proven a determining factor in shape perception [3–6]. In particular, vertical mirror symmetry has proven an important cue to shape extraction from abstract, non-familiar visual elements presented

in conditions of heightened ambiguïty (noise). Across different noise levels, symmetric elements form perceptually more salient shapes than asymmetric ones and are, therefore, more readily detected [5].

The Italian Gestalt psychologist Gaetano Kanizsa [7] discussed a series of ambiguous planar configurations that give rise to powerful figure-ground percepts, with apparent shapes emerging in the foreground, delineated by contours that are perceptually completed beyond physically specified contrast edges. The Gestalt school and Kanizsa himself considered these phenomena as marginal cases of perception ("*margini quasi-percettivi*") and argued that these latter provide insight into the fundamentals of perceptual organization because they put underlying processes to the test under extreme conditions at the capacity limits of the perceptual system. Later-on, the figures seen in such configurations were termed "illusory" by cognitive psychologists; the Gestalt psychologists themselves never used this term, which is, of course, misleading. An illusion, by definition, cannot be verified by independent observation—it only exists in the mind of the person experiencing it. The phenomena described by Kanizsa have clearly defined physical correlates, with measurable systematic effects on perception. One of these configurations is the famous Kanizsa triangle (Figure 1). The Kanizsa figures have been studied extensively to single out factors of physical variation that affect the subjective brightness or darkness of the figures and/or the figure contours. The results from these studies, based on a variety of different experimental measures, are reviewed in sufficient detail elsewhere [8–13]. They are not the object of this study here. Here, we measured the perceptual salience of figure-ground percepts irrespective of the relative darkness, brightness, or clarity of either the induced surfaces or their boundaries, as is made perfectly clear in the instructions given to subjects. As raised previously by others, the response criterion of the subjects in judgment tasks using this type of ambiguous figure [8] is directly dependent on the semantic precision of the instructions given. Formulating these latter appropriately is making sure that the perceptual phenomenon under study, and not a related one, is reflected by the psychophysical data.

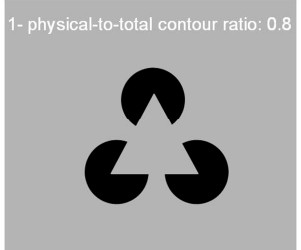

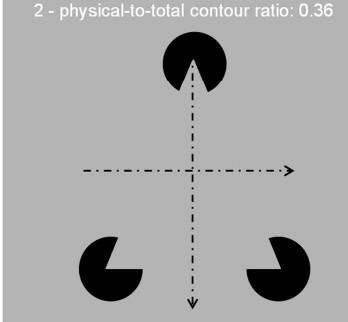 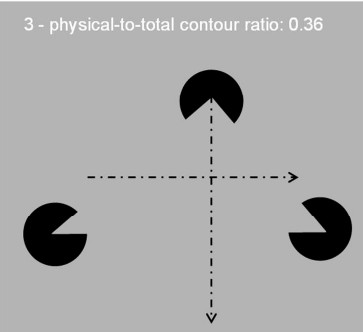

**Figure 1.** Variations of the Kanizsa triangle with phenomenally black inducers on a grey background. The subjective strength of the triangular figure-ground percept emerging in the center of the configuration critically depends on the physically specified-to-total contour, or support ratio. Stronger surface percepts are produced by higher support ratios (**top**). To test for effects of symmetry on the salience of figure-ground, configurations with bilateral symmetry (**bottom left**) and without (**bottom right**) were generated. All the configurations had identical support ratio and, therefore, identical area size. All physically specified elements in the configurations (inducers) were of identical size and contrast intensity.

The influence of variations in the intensity of the local luminance contrast of physical inducers producing the perceptual filling in [14] of bright or dark surfaces, leading to figure-ground percepts at the centre of the configurations in the case of the Kanizsa triangle and similar figures, was demonstrated for the first time by Heinemann's pioneering studies [15]. These were extended later on by others [12] to various configurations including those of the Kanizsa type, where observers had to adjust the luminance of the central figure region until it matched the phenomenal appearance of the general background. This is equivalent to a cancellation of the phenomenal appearance of figure and ground. Configurations produced filling-in consistent with classic simultaneous contrast effects, where a figure appears darker than the general background when it is surrounded by phenomenally white inducers, and brighter when surrounded by phenomenally black inducers. The perceptual salience of such figures increases consistently with the luminance contrast intensity of the inducers, up to some optimal limit. When that optimal limit is reached and the contrast intensity of inducers increases further, the figure intensity is diminished again and may be annulled completely at the highest physical contrast levels [15]. The simultaneous contrast filling-in that leads to the figure-ground percepts in Kanizsa configurations is therefore predicted by specific physical parameters of the inducers. In this study here, these parameters were all controlled experimentally to keep them constant across conditions created to single out an effect of symmetry.

When the physical contrast intensity of the configurational elements is optimal [12–15] and not varied between displays, the next most important physical parameter that straightforwardly determines the subjective strength of the figure-ground percepts in Kanizsa configurations is the physically specified-to-total contour ratio, or support ratio. This was proven in a series of experiments by Shipley and Kellmann [11] using subjective magnitude estimation, a classic psychophysical rating procedure similar to the one applied in this study. Here, the Kanizsa triangle is exploited to probe for effects of symmetry on figure-ground from occlusion cues. The Kanizsa triangle is one of the most cited examples of a specific class of Gestalt configurations where perceived surface depth arises from local occlusion cues. In this specific shape class, figure-ground results directly from a process of surface completion through boundary interpolation across the physically specified edge contours of the inducers providing these local occlusion cues [16–20]. Occluded object completion thus reflects the workings of fundamental visual mechanisms for recovering object percepts from fragmented input, and the ability of human perception to read structure into an apparently chaotic physical world [21,22]. The functional interactions between configurational symmetry and other structural factors in this important perceptual process are still unknown. The dependency of figure-ground salience on the support ratio (Figure 1), a scale-invariant metric, is associated with the ecologically desirable consequence that perceptual salience will not change with variations in viewing distance [11].

At constant physical contrast intensity of configurations with a constant support ratio, the contrast polarity of inducers, i.e., whether they are dark on lighter backgrounds, bright on darker backgrounds or a mixture of both on a medium grey background, does not affect the salience or subjective strength of the resulting figure-ground percept, provided the contrast polarity is homogenous within each of the inducing elements [12,23–29]. When contrast polarity is not homogenous within elements, then, and only then, may the perceptual salience of the figure-ground percept be reduced. This effect may be strongly dependent on the task instructions [8]. Perceptual figure–ground organization is determined by visual mechanisms that integrate contrast intensity and spatial information carried by the configurational elements while mostly discarding information on contrast polarity. This is predicted by sign-invariance models based on functional properties of cortical neurons of the complex type, which are orientation selective but insensitive to contrast polarity [14,23–25,29]. Such sign-invariant visual mechanisms have the ecologically desirable consequence that the simplest plausible representation of figure and ground can be achieved when the signal input from local contrast regions is particularly ambiguous.

The classic version of the Kanizsa triangle is a configuration with perfect vertical mirror symmetry (Figure 1, top, and bottom left). Whether physical display variations producing asymmetric

configurations (Figure 1, bottom right) would affect figure-ground salience in this specific case is not known. The motivation of this study was to test whether symmetry contributes to figure-ground strength in this classic Gestalt configuration, where surface depth results from visual interpolation across fragments. Two variations of the Kanizsa triangle with identical support ratio, as defined by Shipley and Kellmann [11], and identical triangular area size were generated; one with perfect bilateral symmetry (Figure 1, bottom left), the other asymmetric (Figure 1, bottom right). To test for possible interactions between symmetry and the orientation of the configurations in the plane, presentations were varied and bilateral symmetry was not always vertical but could be vertical or horizontal, in a random order. In the light of earlier findings, with abstract shapes presented in noisy contexts [5], vertical mirror symmetry significantly increased the probability that a shape was seen as a figure against the ground. Thus, the bilateral symmetry of vertical orientations may also generate stronger effects on the figure-ground salience of surfaces completed by interpolation. The physical inducers, either dark on grey, light on grey, or light and dark on grey, displayed variations in contrast polarity across inducers, but never within, in both types of configuration, symmetric and asymmetric. In the light of previous findings, these variations should not affect figure-ground strength, given that the polarity of contrast was always homogenous within inducers in the different configurations [8,10,12,26,29].

In a first experiment, psychophysical magnitude estimation was used to measure the salience, or subjective strength, of the figure-ground percepts in the different conditions. In a second experiment, a choice response time was run, with a selected set of configurations, to test whether more salient figure-ground percepts consistently produce, as would be expected, shorter response times with consistent "foreground" response probabilities.

## 2. Materials and Methods

Triangular Kanizsa configurations with and without bilateral symmetry (Figure 1 bottom left and right respectively, for illustration), identical support ratio and surface area, variable orientation (vertical base down, as in Figure 1 bottom left, vertical base up, sideways base left, sideways base right) and variable inducer polarity (three black inducers on grey or '$- - -$', three inducers white on grey or '$+ + +$', two black and one white inducer on grey or '$- - +$', two white and one black inducer on grey or '$+ + -$') were presented in random order to ten human observers in a single-presentation-per-figure subjective magnitude estimation (rating) experiment with *moduli*. Four of these configurations with a single orientation (vertical base down, as in Figure 1 bottom left) and uniformly positive ($+ + +$) or uniformly negative ($- - -$) contrast polarity, two with vertical symmetry and two without, were presented to six additional human observers in a repeated measures (four presentations per configuration) choice response time experiment with two additional control stimuli (triangles with minimally visible line contours ("ghosts") only, no surface contrast).

### 2.1. Stimuli

The image configurations were computer generated using an HP Zbook 15 G2 Mobile Workstation equipped with a 4th generation Intel Core i7-6700 processor and an NVIDIA Quadro K5100 graphic card.

Configurational dimensions in terms of size (in pixels and in cm on screen) of triangle base and triangle sides, the inducer radius, which was identical in all the configurations, and the overall physical-to-total contour ratio, also identical in all the configurations, are summarized in Table 1. Luminance values of the different configural elements were determined photometrically using an OPTICAL photometer (Cambridge Research Systems). The ADOBE RGB coordinates of the phenomenally grey background (RGB: 140, 140, 140) yield a background luminance of 55 cd/m$^2$. The phenomenally black inducers (RGB: 5, 5, 5) a luminance of 4 cd/m$^2$, and the phenomenally white inducers (RGB: 240, 240, 240) a luminance of 98 cd/m$^2$. The *moduli* from the subjective rating task (RGB: 135, 135, 135 for the phenomenally darker ones and RGB: 145, 145, 145 for the phenomenally lighter ones) had a luminance of 52 cd/m$^2$ or 58 cd/m$^2$ respectively. The line contour control configurations

(RGB: 120, 120, 120) from the choice response time task had a luminance of 48 cd/m$^2$. The physically specified contrast intensities with positive and negative signs may be calculated using the Weber Contrast (Weber Ratio, *W*) formula:

$$W = (L_{config} - L_{background})/L_{background} \tag{1}$$

**Table 1.** Figure dimensions in centimeters (cm) with the overall support ratio and surface area as a function of configuration (*symmetric versus asymmetric*). The symmetry factor only varies systematically between configurations, the shape interpretation ("triangle") is the same and so are all relevant physical parameters.

|  | *Symmetric* | *Asymmetric* |
| --- | --- | --- |
| *Triangle base (b)* | 9 cm | 13 cm |
| *Triangle side 1* | 12 cm | 11 cm |
| *Triangle side 2* | 12 cm | 9 cm |
| *Triangle height (h)* | 11 cm | 7.62 cm |
| *Triangle surface area (1/2bxh)* | 49.5 cm | 49.5 cm |
| *Physical inducer radius* | 2 cm | 2 cm |
| *Support ratio* | 0.36 | 0.36 |

As a consequence, we have a positive *W* of +0.92 for the phenomenally white inducers, a negative *W* of −0.78 for the phenomenally black inducers, a positive and a negative *W* of +0.09 and −0.09 respectively for the minimal-contrast *moduli* from the subjective rating task, and a negative *W* of −0.13 for the minimal contrast line contour control configurations from the choice response time experiment.

*2.2. Presentation of Configurations*

The configurations were presented in random order on the screen of the HP Zbook 15 G2 Mobile Workstation, which has a pixel resolution of 1920 × 1080 and a 60 Hz refresh rate. Random selection, presentation, and response coding were computer controlled using Python for Windows. The duration of presentation of each single configuration was observer controlled in both experiments, a subsequent presentation always initiated 800 milliseconds after the observer had typed his/her response on the computer keyboard. The 32 configurations from the single-presentation subjective magnitude estimation (rating) task, with the different variations in orientation and in local contrast polarity, are shown in Figure 2a,b, for illustration. Illustrations of the 6 configurations from the repeated measures choice response time task are shown in Figure 3.

*2.3. Experimental Procedure*

Subjects were seated in front of the workstation at a distance of about 90 cm from the screen in a semi-dark room. In the subjective magnitude estimation task, they were shown a set of *moduli* consisting of minimal-contrast triangular surfaces of the same spatial dimensions as the symmetric and asymmetric triangular centers of the test configurations. These *moduli* are shown in Figure 2c for illustration. Subjects were told to associate the phenomenal strength of the *moduli* with a rating score of '11'. It was then made clear to them that they would be shown different triangular configurations, with black and white patches around them. They were then asked to rate the subjective strength of the figure-ground percept at the centre of the test configurations, in terms of the strength of the segregation into foreground and background, regardless of the direction of the perceived contrast (i.e., subjectively darker or subjectively lighter), by a number between '0' and '10', bearing in mind that the highest number was to reflect a figure salience closest to that of the real-contrast *moduli* they had seen just before. Each of the 16 asymmetric and the 16 symmetric configurations (Figure 2a,b, respectively, for illustration) was presented only once to each of the subjects in a single random order session. In the choice response time task, subjects were asked to judge as swiftly as possible

whether the triangle displayed on the screen seemed to stand out as foreground against the grey general background, or to lie behind the general grey background. In this experiment, the outlined triangular shape control configurations (Figure 3, on the right) were presented at a minimal, just visible negative line contrast intensity and no surface contrast. This renders a highly ambiguous (one subject mentioned "ghost-like") appearance on the screen with no clear figure-ground assignment. Each of the six configurations (Figure 3, for illustration) was shown four times, in random order, to each of six subjects in a single individual session.

## 2.4. Subjects

Ten individuals (six men, four women) between 20 and 31 years old, all of them with normal or corrected to normal vision, participated in the subjective magnitude estimation experiment. Six further individuals (five men, one woman), also young and with normal or corrected to normal vision, participated in the choice response time experiment. Participants were mostly undergraduates involved in medical or language studies. None of them was familiar with the configurations presented to them, and all of them were naïve to the purpose of the study. The experiments were conducted in accordance with the Declaration of Helsinki (1964) and in full conformity with the author's host institution's (CNRS) ethical standards committee. Informed consent was obtained from each of the participants.

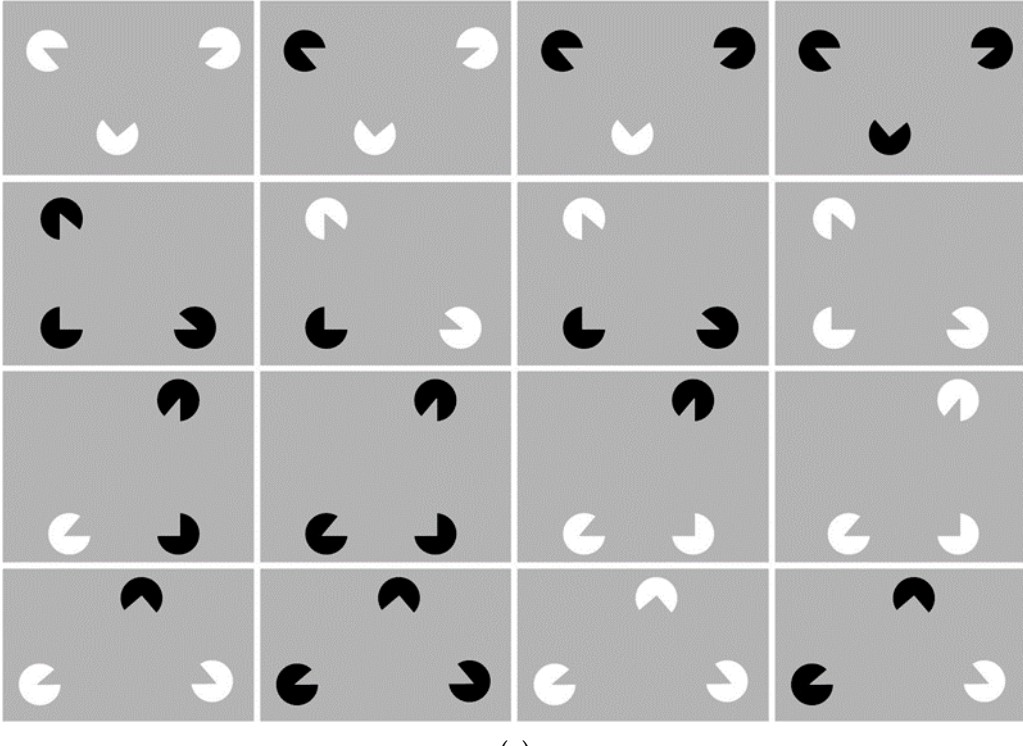

(**a**)

**Figure 2.** *Cont.*

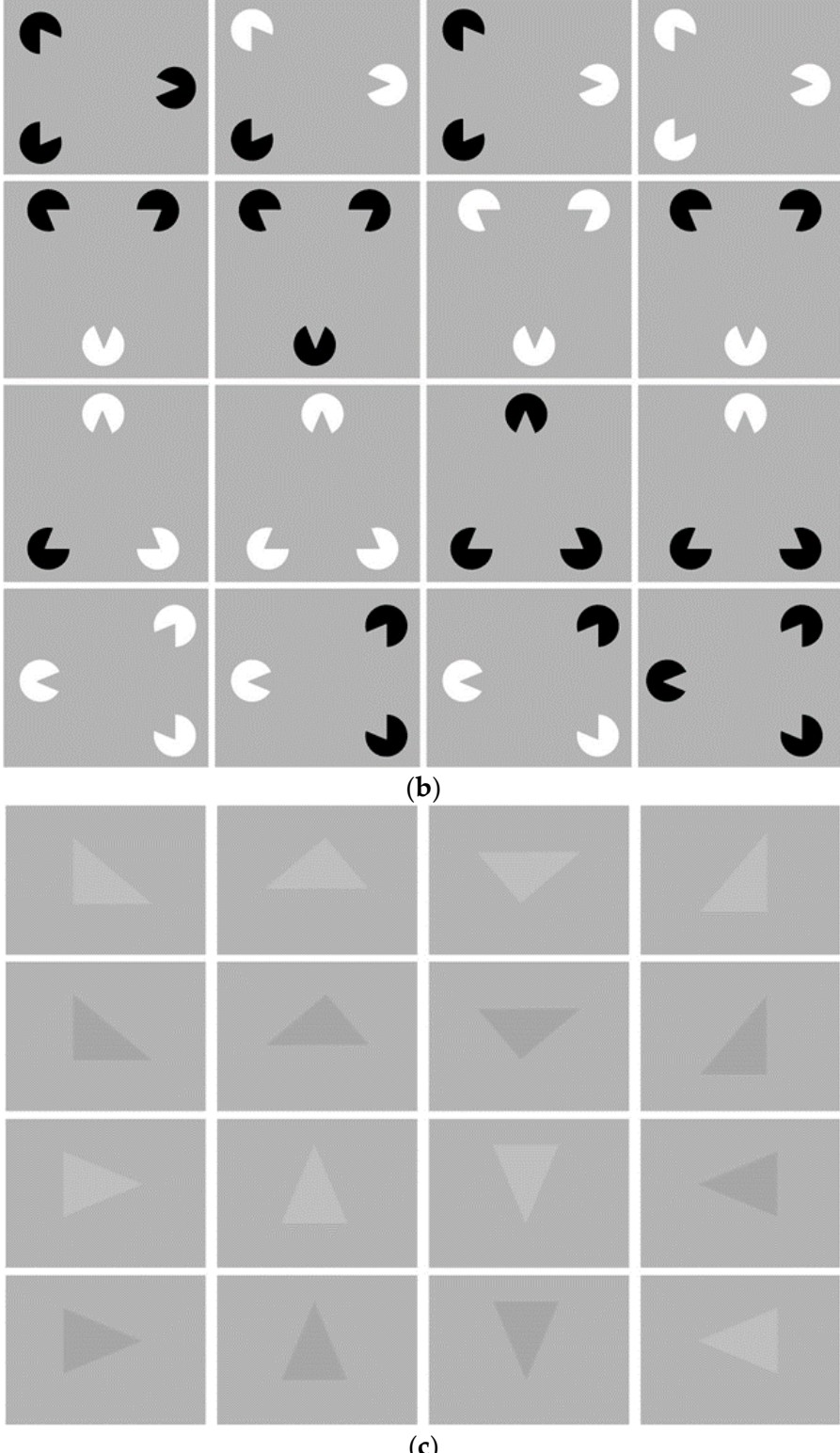

(b)

(c)

**Figure 2.** The configurations from the subjective rating experiment (**a**) The asymmetric Kanizsa triangles, with varying orientation and inducer contrast polarities; (**b**) The Kanizsa triangles with axial symmetry; (**c**) the *moduli* for benchmarking the subjective rating scale (0–10). Subjects were told to associate the *moduli* with a figure-ground strength rating of '11', regardless of the direction of the perceived contrast.

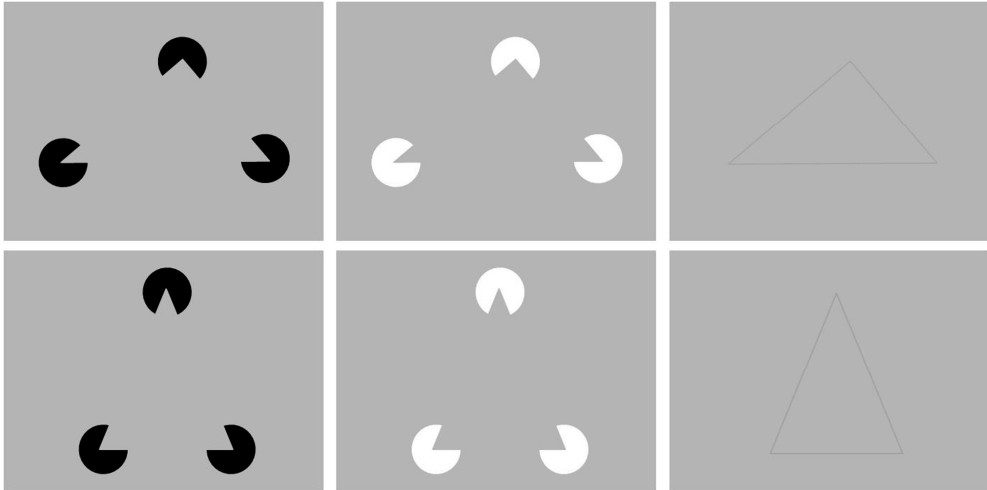

**Figure 3.** The test and control configurations ("ghosts") from the choice response time experiment. Six subjects were asked to judge as swiftly as possible whether the triangle displayed on the screen seemed to stand out as foreground against the grey general background, or to lie behind the general grey background.

### 2.5. Data Analysis

The data from the subjective rating experiment, with a Cartesian design plan written in terms of $Subject_{10} \times Symmetry_2 \times Orientation_4 \times Polarity_4$, produced a total of 320 subjective ratings. These data were fed into a Three-Way ANOVA. Means, standard errors, effect sizes, and F statistics with probability limits were determined.

The data from the choice response time experiment, with a Cartesian design plan written in terms of $Subject_6 \times Symmetry_2 \times Polarity_3 \times RepeatedMeasures_4$, produced a total of 144 choice data and a total of 144 response times. In the experimental design plan, the control configuration represents the third modality of the "polarity" factor, with the three factor levels "positive" or '+ + +', "negative" or '− − −', and "control". The response times were fed into a Two-Way Repeated Measures ANOVA with individual data averaged over the four levels of the repetition factor $R_4$ and without the third level of the "polarity" factor, i.e., the analysis plan therefore reads $Subject_6 \times Symmetry_2 \times Polarity_2$.

## 3. Results

The results of the analyses on the data from the subjective rating experiment and the choice response time experiment are shown here below in the form of graphs and tables.

### 3.1. Subjective Magnitude Estimation Task

The individual data from this experiment are made available in Table S1 of the Supplementary Materials section. The data show a good consistency between participants, within and across conditions, with a moderate amount of inter-individual variability. The main effect of symmetry on the subjective magnitude of the figure-ground percept in the Kanizsa configuration is highlighted by the two graphs in Figure 4. The average magnitude of figure salience in terms of average subjective ratings produced by symmetric and asymmetric configurations is plotted as a function of the orientation of the configurations in the plane (Figure 4, top), and as a function of the contrast polarity of the inducers (Figure 4, bottom). While the subjective ratings display no major variations with either orientation of the configurations or the contrast polarity of the inducers, they are consistently and noticeably affected by lack of symmetry. Subjective ratings are found to be markedly stronger for all the configurations with bilateral symmetry, irrespective of orientation and/or contrast polarity. This effect is highlighted further by the statistics shown in Table 2 here below, which summarizes the observations from Figure 4

in terms of results from the Three-Way ANOVA on the subjective ratings of the ten subjects. The effect of symmetry is statistically significant, the effects of orientation and inducer polarity are not.

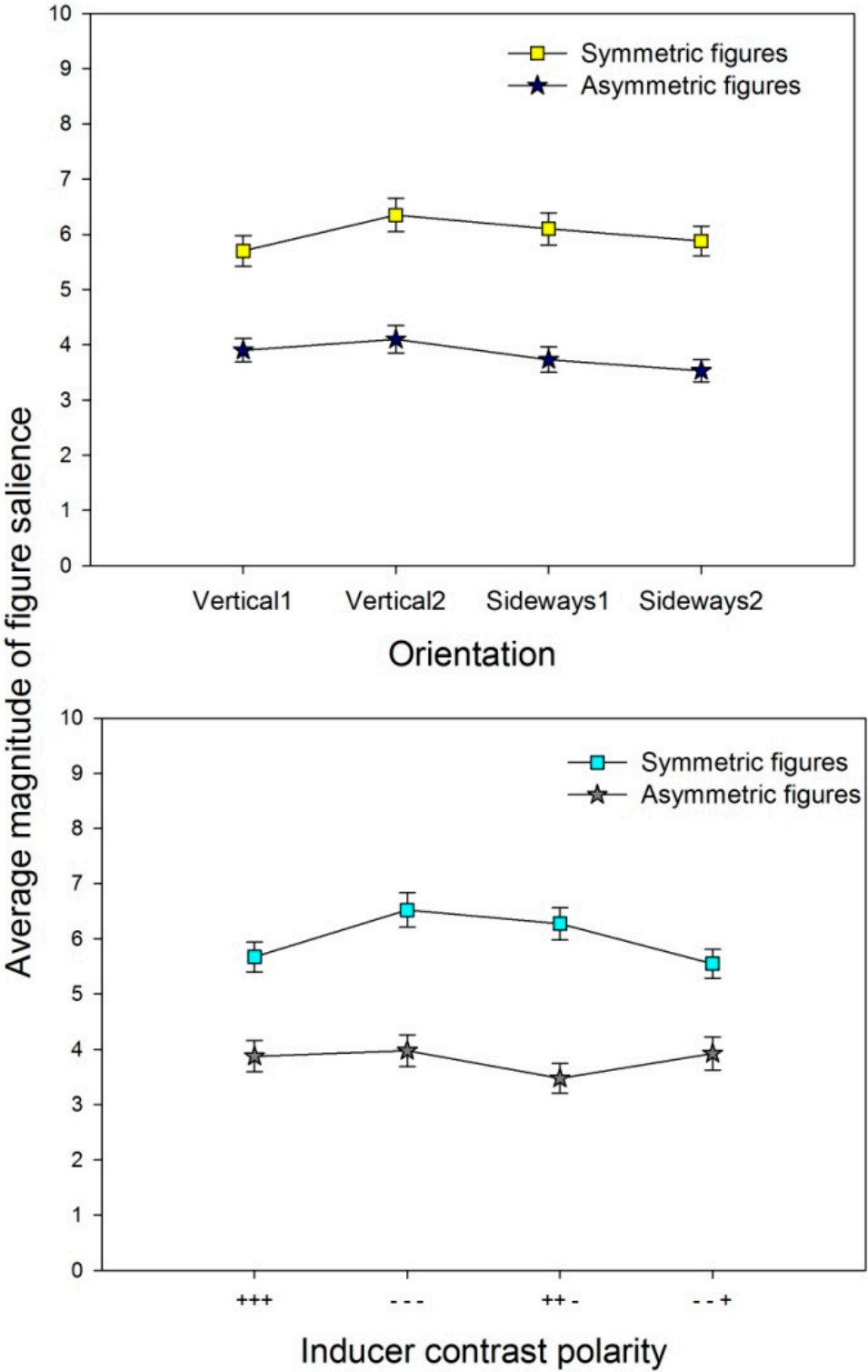

**Figure 4.** Average magnitudes of figure-ground in terms of average subjective ratings with bars indicating +/− the standard error of the mean. Effects produced by symmetric and asymmetric configurations are plotted as a function of the orientation of configurations in the plane (**top**), and as a function of inducer polarity (**bottom**).

**Table 2.** Three-Way ANOVA results with the means (average subjective magnitudes), standard errors (SEM), and *F* statistics for effects of main factors and their interactions from the analysis of the subjective rating data.

| Factor | Level | Mean | SEM | F |
|---|---|---|---|---|
| Symmetry ($S_2$) | asymmetric | 3.8 | 0.15 | $F(1, 319) = 108.8; p < 0.001$ |
|  | symmetric | 6.1 | 0.13 |  |
| Polarity ($P_4$) | − − − | 4.7 | 0.21 |  |
|  | + + + | 5.3 | 0.22 | $F(3, 319) = 1.24$; NS |
|  | − − + | 4.8 | 0.20 |  |
|  | + + − | 4.7 | 0.19 |  |
| Orientation ($O_4$) | vertical base bottom | 4.8 | 0.20 |  |
|  | vertical base top | 5.2 | 0.23 | $F(3, 319) = 1.17$; NS |
|  | sideways base left | 4.9 | 0.21 |  |
|  | sideways base right | 4.7 | 0.22 |  |
| Symmetry × Polarity | interaction | _ | _ | $F(3, 319) = 1.83$; NS |
| Symmetry × Orientation | interaction | _ | _ | $F(3, 319) = 0.41$; NS |
| Polarity × Orientation | interaction | _ | _ | $F(9, 319) = 0.67$; NS |

### 3.2. Choice Response Time Task

The raw data from this experiment are made available in Table S2 of the Supplementary Materials section. The perceptual judgments from the choice task, shown here below in Table 3 in terms of the percentage of "foreground" responses as a function of configuration (*asymmetric vs. symmetric*) and the contrast polarity of the inducers (*negative vs. positive vs. control*), expressed in terms of the phenomenal appearance of the inducers here, display a consistent majority of "foreground" responses in the ambiguous Kanizsa configurations, with noticeably higher percentages of "foreground" in the configurations with bilateral symmetry, irrespective of inducer appearance (or polarity). In response to the control configurations, the percentages of "foreground" responses show no such clear trend. This is explained by the fact that the outlined shape control configurations (Figure 3, on the right) were presented at a minimal, just visible negative line contrast intensity, which made them particularly ambiguous with respect to figure-ground organization in the plane. The outlines are not perceived as clearly belonging to a specific depth level, which is reflected in the results here by a near random distribution of "foreground" and "background" responses. This suggests that the outlined controls without surface contrast did not produce, as could be expected, salient figure-ground percepts.

**Table 3.** Percentage of "foreground" responses from the choice response time task as a function of configuration and inducer contrast polarity.

|  | Asymmetric | Symmetric |
|---|---|---|
| White inducers | 88% | 98% |
| Black inducers | 75% | 92% |
| Control | 70% | 55% |

Average choice response times were plotted as a function of configuration and inducer polarity, as shown here in Figure 5. The graphs show a consistent and systematic effect of symmetry on response times. Subjects respond markedly faster to configurations with axial symmetry, irrespective of whether the inducers are phenomenally black (negative contrast polarity) or white (positive contrast polarity). This effect is highlighted further by the statistics shown in Table 4 here below, which summarizes the observations from Figure 5 in terms of results from the Two-Way Repeated Measures ANOVA on the response times of the six subjects. The effect of symmetry is statistically significant, the effect of inducer polarity is not. The third level of the "polarity" factor here, i.e., the control configuration, was not included in the design plan for this ANOVA.

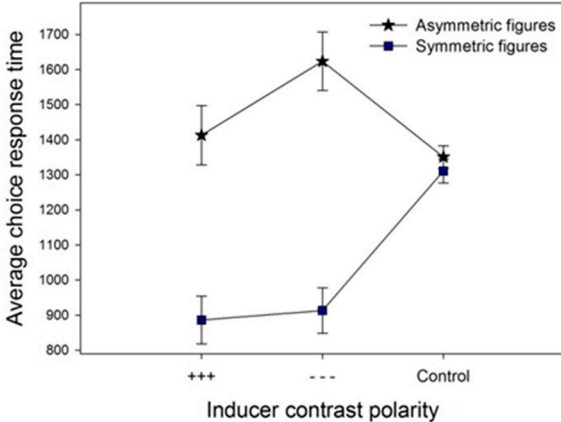

**Figure 5.** Average choice response times with bars indicating $+/-$ the standard error of the mean as a function of configuration and inducer polarity.

**Table 4.** Two-Way Repeated Measures ANOVA results with the means (in milliseconds), standard errors (SEM), and *F* statistics for effects of main factors and their interactions from the analysis of the choice response times.

| Factor | Level | Mean | SEM | F |
|---|---|---|---|---|
| Symmetry ($S_2$) | asymmetric | 1518 | 73 | $F(1, 23) = 36.69; p < 0.01$ |
| | symmetric | 900 | 65 | |
| Polarity ($P_2$) | $- - -$ | 1269 | 64 | $F(1, 23) = 1.74; NS$ |
| | $+ + +$ | 1150 | 62 | |
| Symmetry × Polarity | interaction | – | – | $F(1, 23) = 1.21; NS$ |

## 4. Discussion

Although symmetry has been discussed in terms of a major grouping principle or law of good Gestalt since Wertheimer and Metzger [1,2], the specific effects symmetry may produce on feature grouping, figure-ground segregation, visual discrimination, or time to respond to visual configurations have become subject to systematic quantitative investigation in perceptual science only recently. In the case of visual perception, symmetry may be conceived as a geometric property that yields configurational simplicity and, therefore, represents an ecological advantage for information processing [30]. Symmetry may also be seen as a perceptual feature that attracts attention, enhances configural salience, and facilitates grouping [5,31–35].

It is found that bilateral configurational symmetry, i.e., mirror symmetry within the whole configuration, strengthens the perceptual salience of figure against ground in triangular Kanizsa configurations. The results from the magnitude estimation (rating) experiment clearly show that the subjective strength of the foreground at the center of the configurations is significantly higher when the configurations have bilateral symmetry. This holds for triangular configurations when mirror symmetric configurations are compared to asymmetric configurations with the same number of inducers, and the same support ratio as defined by Shipley and Kellman [11]. This symmetry effect can be exploited to further quantify critical interactions between occlusion-based surface properties, symmetry, and figure-ground salience. Variations in symmetry could be tested against variations in the support ratio in the first instance. Figure-ground segregation from interpolation is an early-stage process in perceptual grouping [16,36–45], as is symmetry detection [39,43]. In particular, as shown by Erlikhman and Kellmann [44,45], the human perceptual system uses critical spatial cues of local position and alignment within a restricted spatiotemporal window (~165 msec) for the rapid extraction of co-oriented edge fragments from the visual input. As predicted by the Gestalt Law of Good Continuation, the fragments then connect by known neural interpolation processes [16,25,35], producing larger surfaces

that will stand out as figures against ground. The results from the experiments here show that symmetry contributes to this early process of perceptual organization.

The results suggest no influence of the orientation (vertical versus horizontal) of the axis of symmetry on the salience of the figure-ground percepts. Vertical and horizontal mirror symmetry produced equally strong phenomenal salience of figure-ground. Since Mach [33], it is suggested that symmetry around the vertical axis may be more effectively processed by the visual system than symmetry about the horizontal, or any other, axis in the plane. Some studies have confirmed this prediction [5]. However, more recent reviews indicate that there may be no systematic functional advantage of vertical symmetry [34]. Effects of axis of symmetry on perception may be dependent on what Bertamini termed "objectness" [31], i.e., whether the cognitive interpretation of the visual shape changes with translational or rotational changes of the latter. Psychophysical data on shape perception [43], using radial frequency patterns and other objects, indeed suggest that variations in the location and orientation of relevant (with respect to the perceptual task) object features may generate effects of axis of symmetry. In the two experiments here, relevant perceptual features within and across objects (*symmetric vs. asymmetric*) can be considered invariant, since there was no effect of contrast polarity and no interaction between contrast polarity and symmetry. This could explain why the axis of symmetry had no effect here either. Also, earlier psychophysical studies have shown that bilateral symmetry is significantly more salient within objects, significantly less between objects [39,40]. The layout characteristics, including symmetry, of complex figure-ground solutions are more easily processed within single perceptual objects [40]. Symmetry detection becomes harder with complex shape configurations where other factors, such as positional uncertainty or convexity, interact with the symmetry factor, especially when the psychophysical task requires comparing across objects. It may be that vertical symmetry generates a measurable advantage for perception only under such conditions.

The results from the choice response time task show a consistently higher percentage of "foreground" responses to the symmetric configurations, which is accompanied by significantly shorter response times. Bilateral symmetry, therefore, represents a measurable functional advantage in the perceptual processing of figure-ground. Variations in the contrast polarity of the inducing elements had no marked effect on either the subjective strength of the figure-ground percepts or the response times. This observation is consistent with results from earlier work with similar configurations where symmetry was not varied [9,12,29], and predicted by non-linear neural models of figure-ground based on the long-range integration of antagonistic brightness signals [12–14,23,24]. Interestingly, when inducers of both positive and negative contrast signs, i.e., phenomenally white and black inducers, are present in the same configuration, the latter may be perceived as phenomenally asymmetric with respect to brightness. The perceptual system, however, is not influenced by the symmetry/asymmetry in contrast signals, only by geometrically defined symmetry. Although this study here was not specifically aimed at singling out the hierarchal level of perceptual processing at which the symmetry effect arises, it is unlikely that conscious processing was involved here. After the experiments, subjects were asked whether they had noticed anything in particular in the configurations or used a particular strategy to respond. Most of them stated that "some were the same, some were different", or "some had white parts, some had black parts, some had both", but none of them was able to specify any particular structural difference or response strategy. We may therefore infer that subjects were not immediately aware of the systematic variations in symmetry.

Bilateral symmetry is identified as a key prior for three-dimensional shape perception in humans [41,42]. The perceptual integration of symmetry in this process does not necessarily happen consciously and, as explained above, may vary with shape complexity and shape interpretation [40] without subjects being aware of it. Therefore, configurational complexity and shape interpretation need to be controlled for to single out the effects of symmetry *per se* on any particular aspect of perceptual organization. This is clarified further by some of the additional illustrations in Figure 6, showing configurations where the manipulation of symmetry inevitably implies changing also the complexity of the configuration as a whole, and the resulting shape interpretation. In the square version of the Kanizsa figure, breaking the configurational symmetry inevitably requires changing the shape borders.

This produces a new, far more complex, qualitatively different shape geometry leading to a radically different shape interpretation. The Kanizsa square is therefore ill-suited for singling out effects of symmetry without ambivalence. The advantage of the triangular configuration used in this study is that the geometric transformations needed to manipulate mirror symmetry affect neither the structural complexity of the configurations, nor the resulting shape interpretation: with or without bilateral symmetry, the perceptual solution is always and only a "triangle".

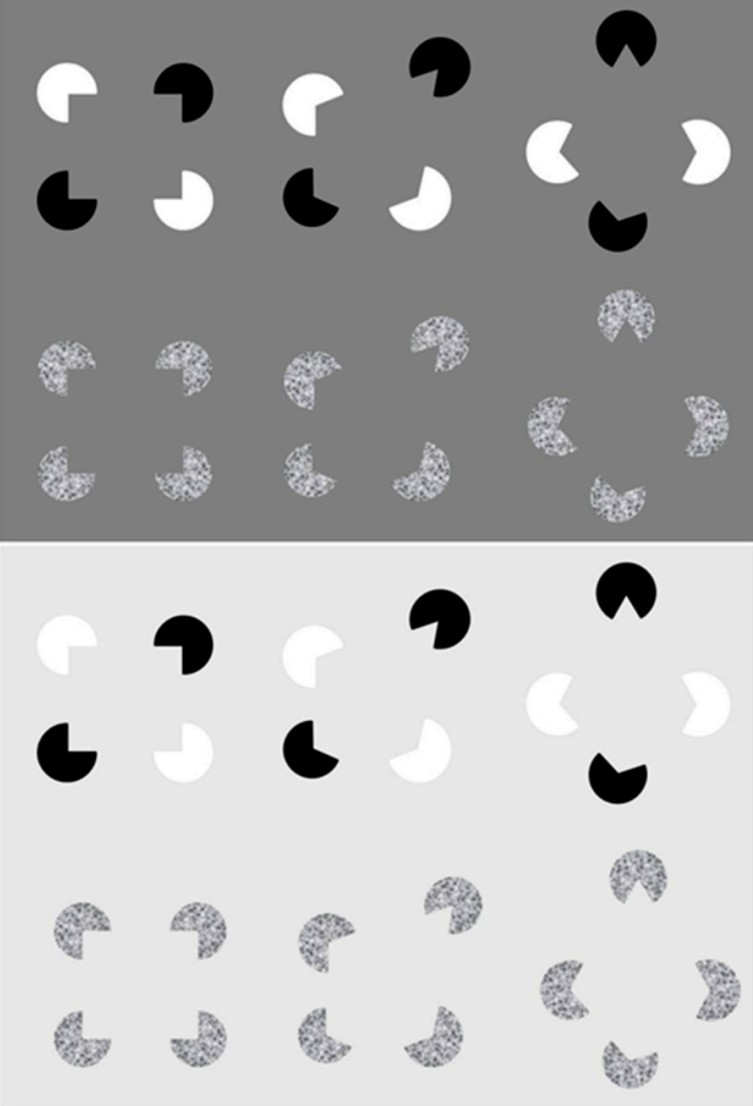

**Figure 6.** An example of a configuration where effects of symmetry on visual perception cannot be tested independently from possible effects of shape interpretation. In the square version of the Kanizsa figure, breaking the symmetry of the classic square configuration (images on left) inevitably requires breaking the perpendicularity of the shape borders. This inevitably results in a new and qualitatively different shape geometry and shape interpretation. In other words, shape interpretation then becomes a confusion factor. In this case here a "shard" with qualitatively different 3D-like shape properties emerges (images in middle and on left). When shape orientation changes, the percept changes, again, qualitatively and a new visual object emerges (cf. on the importance of "objectness", see Bertamini [31]). All configurations here above have roughly equivalent, albeit not strictly identical, support ratio and central area size. Variations in inducer texture, figure orientation, and background intensity are presented here for illustration only.

The new effect found here is fully consistent with the adaptive logic of visual preference for symmetry, where symmetry strengthens the figure-ground salience of surfaces from occlusion cues, formed through visual spatial interpolation across fragments within a narrow temporal window of processing [44,45]. Symmetry strongly influences perception-based decision making in humans and in many animals, and survival-relevant responses to symmetry, or lack thereof, are found not only in primates but also in other species [37]. This highlights the wider biological significance of symmetry as a visual signal, with further implications for higher order adaptive human behavior, such as structural design [46], or image-guided precision tasks [47,48]. The early Gestalt theories intuitively captured this fundamental importance in a large body of observations on phenomena of human perceptual organization. Their intuitions were astute, pointing towards functional aspects of symmetry which perception science has only just begun to quantify and predict.

**Supplementary Materials:** The following are available online at http://www.mdpi.com/2073-8994/11/2/225/s1, Table S1: Subjective Magnitude Estimation Experiment Data, Table S2: Choice Response Times Experiment Data.

**Funding:** This research received no external funding.

**Acknowledgments:** The experiments were materially supported by the CNRS. The contribution of Cyrielle Giger-Dechavanne, who helped recruiting and running subjects, is most gratefully acknowledged.

**Conflicts of Interest:** The author declares no conflict of interest.

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
