# Peer review of "Bilateral Symmetry Strengthens the Perceptual Salience of Figure against Ground"

_symmetry, doi:10.3390/sym11020225_

Round 1
Reviewer 1 Report
This article describes a study using ambiguous Kanizsa configurations. Fourteen observers reported "foreground" or "background" percepts. The aim was to test the effect of symmetry on the strength of figure-ground organisation. Symmetrical configurations produced greater magnitudes and higher percentages of "foreground" judgments.
In my opinion there is a better way the pose the question. This study does not seem to test the effect of symmetry on the subjective strength of figure-ground, but rather the effect of symmetry on the strength of an illusory surface. Illusory surfaces are affected by a number of factors, as the author describes in the intro, and one of them is spatial configuration. I think it may be correct to say that regularity (symmetry) has not been tested systematically, although it has been discussed at length, including by Kanizsa. But this is a very special case of figure-ground, and there is no reason to assume that illusory surfaces are a good model for figure-ground in general.
A focus on illusory surfaces may mean a change in the structure of the paper. In case the paper were to retain a focus on symmetry and grouping, there is also a relevant literature on how bilateral symmetry is more salient within rather than between objects: Baylis & Driver, 2001, Visual Cognition; Bertamini, 2010, Perception; Makin et al., 2014, JOV.
Task. I am puzzled by the speeded task described as "In the choice response time task, subjects were asked to judge as swiftly as possible whether the triangle displayed on the screen seemed to stand out as foreground against the grey general background, or to lie behind the general grey background."
Why would the triangle appear to lie behind the background? Maybe there are problems with the figures but this description would make no sense to me as a naive observer.
Debrief. This kind of task may suffer from demand characteristics. Observers may notice the symmetry and decide to take it into account. This needs discussion. I strongly recommend that in such cases observers are carefully interviewed after the task to ask what they thought about the study, about its aim, and whether they adopted any conscious strategy.
Figure 2. Something went wrong with the figures in the pdf file and some shapes are not visible.
Line 34 " In physcial science" Typo
Author Response
The configurations, experimental conditions, and task instructions were designed to study figure-ground salience resulting from structurally ambiguous configurations, not the subjective strength of the (often so-called) "illusory" surfaces.
The subjective strength of the induced triangles per se is predicted constant when keeping inducer size and contrasts, interpolation support ratio (Shipley and Kellman, cited) and induced surface size constant. The interpolation support ratio was not varied in the triangular Kanizsa configurations, the size of the induced triangular surface was also invariant, please see Table 1, as is explained in the introductory and in the materials-and-methods sections. Some additional and detailed explanations have been added to clarify this even further in the revised manuscript, right at the beginning, on page 2. All changes made are highlighted in red in the new text.
The Gestalt school and Kanizsa himself considered the figure-ground phenomena in the type of configuration used in my study as limiting cases of perception ("margini quasi-percettivi"), and argued that these later are well-suited to provide insight into the fundamentals of perceptual organization, by putting underlying processes to the test under extreme conditions, at the perceptual system's capacity limits. Later-on, the figures that can be seen in such configurations were termed "illusory" by cognitive psychologists (1960 and thereafter); the Gestalt psychologists themselves never used the term "illusory", as it is, of course, misleading. An illusion, by definition, cannot be verified by independent observation - it only exists in the mind of the person experiencing it. The phenomena described by Kanizsa have clearly defined physical correlates, with measurable systematic effects on perception. One of these configurations is the famous Kanizsa triangle. This and other Kanizsa figures have been studied extensively to single out factors of physical variation that affect the subjective brightness or darkness of the induced figures and/or the figure contours. The results from these studies are reviewed in sufficient detail elsewhere [8-13], and are not the object of this study here.
Here, we measure the perceptual salience, or perceived strength, of the induced figure-ground percept irrespective of the relative darkness, brightness, or clarity of either the indced surfaces or their boundaries, as is made perfectly clear in the instructions given to subjects. As raised previously by others (Victor and Conte, cited), the response or task criterion of subjects when presenting them with this type of ambiguous figure is directly dependent on the semantic precision of instructions given. Formulating these latter appropriately is to make sure the perceptual phenomenon under study, not a related one, is reflected by the psychophysical data. In this study here, the precise task instructions were to "rate the strength of the figure-ground phenomenon at the centre of the configuration, in terms of a segregation into foreground and background, regardless of the direction of the perceived surface contrast."
3- Nowhere in the manuscript is it stated that "illusory surfaces" would be "good models for figure-ground in general"! The term "illusory" is not even mentioned a single time in the manuscript, for reasons given here above. The manuscript adheres strictly to the nomenclature of the Gestalt psychologists, invoking limiting cases of perceptual organization to bring to the fore structural laws of figure and ground. Also, to take this very interesting debate a little further and to highlight the reasons why a triangular configuration was chosen here even further, I have added examples illustrating the impossibility to separate variations in symmetry from variations in shape complexity and shape interpretation (cf. "objectness" in the terminology of Bertamini, 2010) in the square-shaped Kanizsa configurations. Please see the new text on pages 11-13 of the revised manuscript, with an additional figure for illustration. This also helps to make clear why the triangular version of this configuration was chosen to single out effects of bilateral symmetry most adequately.
The suggested results from previous work showing that bilateral symmetry is more salient within rather than between objects have been incorporated into the revised discussion.
The outlined control triangles in the control configurations had minimally perceptible boundary contrast on the screen and no surface contrast (they may look stronger in the figures, provided for illustration, in the manuscript). When seen without context on a grey background, they have, indeed, no clear assignment to "figure" or "ground" - be they symmetrical or not (this makes them perfect controls against configurations with figure-ground assignment). Two subjects mentioned that these outlined shapes looked like "ghosts" in "grey fog". The likelihoods that they are seen as "background" or "foreground" were, indeed, not expected to differ significantly from the chance level.
We did (and always do) ask the subjects after the experiment whether they noticed anything particular, or used a strategy to respond. This is now explicitly mentioned and discussed on page 13 of the revised manuscript (new discussion): "Although this study here was not specifically aimed at singling out the hierarchical level of perceptual processing at which the symmetry effect arises, it is unlikely that conscious processing was involved here. After the experiments, subjects were asked whether they had noticed anything in particular in the configurations, and most of them stated that "some were the same, some were different", or "some had black parts, other white parts, some had both", but none of them was able to specify any particular structural difference, a particular cue they may have used for responding". We may therefore infer that subjects were not aware of the variation in symmetry."
Minor:
The problem with this Figure is specific to the pdf version of the manuscript, please upload the doc version for clearer figures. I have also sent originals to be made available with the revised manscript.
The typo was corrected. Many thanks for your time and interesting comments and recommendations.
Reviewer 2 Report
In the manuscript, titled "Axial symmetry strengthens the perceptual salience of figure against ground", by Birgitta Dresp-Langley, the author compared perception of Kanizsa triangle with mirror-symmetrical and asymmetrical shapes of the triangle in two behavioral experiments. Participants rated subjective strength (or saliency) of the illusory triangle in the 1st experiment and responded whether the illusory triangle was foreground or background in the 2nd experiment. Results of the experiments show that the subjective strength became higher in the 1st experiment and the triangle was more frequently judged as foreground with faster reaction time in the 2nd experiment with the symmetrical shape than with the asymmetrical shape. Unfortunately, the experiments were not well designed and it is impossible to make any inference/conclusion based on the results reported in this manuscript.
The author used only a single pair of symmetrical and asymmetrical shapes of the triangle in this study. It is impossible to discuss any role of symmetry based on only these two samples of the triangle shape. There are some other properties of 2D shapes that can affect perception of the shapes (e.g. 2D compactness, near-perpendicularity (80.4 deg) of the top angle of the asymmetrical triangle). Difference of the results between conditions with the symmetrical and asymmetrical shapes could be explained by these shape properties (or by difference of size of the triangle between the symmetry and asymmetry conditions). Note that the author did not observer any effect of an orientation of a symmetry axis in this study but it has been shown that the orientation of the symmetry axis plays an important role for visual perception in many prior studies. Potentially, the difference between this study and the prior studies can be explained by the limitation of visual stimuli used in this study.
Configuration of inducers (a.k.a. packmans) for the illusory triangle was mirror-symmetrical for the symmetrical triangle shape but not for the asymmetrical triangle shape in the experiments. Any difference of perception between conditions with the symmetrical and asymmetrical shapes could be explained by the symmetrical configuration of packmans rather than symmetrical shape of the illusory triangle. Note that the packman configuration can be asymmetrical while the illusory triangle induced by them is still symmetrical. So, these two factors can be individually controlled.
I also have some concerns about statistics in this study. The results were analyzed using multi-way ANOVA (repeated-measures) in the both experiments but the statistical results reported in the manuscript are incomplete. The author does not report any statistical result of interactions (Tables 2 and 4).
Minor issues:
There are some inconsistencies of descriptions about statistical method used in this study. The main text of the manuscript in the 2nd experiment says “Two-Way Repeated Measures ANOVA…reads Subject 6 x Symmetry 2 x Polarity 2 x RepeatedMeasures 4”. In this case, the degrees of freedom of error term (df2) should be 95 (6x2x2x4 – 1) but df2 in Table 4 is 23. Also, the text says “Two-Way Repeated Measures ANOVA” but caption of Table 4 says “Three-Way ANOVA”.
The author should show results of the experiments with graphs properly designed. In Figures 4 and 5, the author connected plots with contours. It implicitly means that the author interpolates between conditions but these conditions are hardly interpolated. The conditions represent polarity of luminance and symmetry/asymmetry of the triangle shape.
I suspect Equation 1 should be written as: W = (Lconfig – Lbackground) / Lbackground. The equation needs a pair of parentheses in its numerator above Lbackground. (Otherwise, Lconfig – Lbackground / Lbackground = Lconfig – 1.)
Author Response
Subjects were not asked to rate the subjective strength of the "illusory" triangle in the rating task. The precise task instructions were to "rate the strength of the figure-ground phenomenon at the centre of the configurations, in terms of a segregation into foreground and background, regardless of the direction of the perceived surface contrast." The term "illusory" is not even mentioned a single time in the manuscript, which adheres strictly to the nomenclature of the Gestalt psychologists, invoking limiting cases of perceptual organization to bring to the fore structural laws of figure and ground. The subjective strength of the induced triangles and their subjective boundaries per se is predicted constant when keeping inducer size and contrasts, interpolation support ratio (Shipley and Kellman, cited) and induced surface size constant. The interpolation support ratio was not varied in the triangular Kanizsa configurations here, the size of the induced triangular surface was also invariant, please see Table 1, as is explained in the introductory and in the materials-and-methods sections. Some additional and detailed explanations have been added to clarify this even further in the revised manuscript, right at the beginning, on page 2. All changes made are highlighted in red in the new text.
Thus, to state the "experiments were not well designed" is unfair and unfounded. As now clarified further in the revised manuscript (all new text is highlighted in red for easy tracking), the triangular configurations were chosen specifically to single out the effects of bilateral or mirror symmetry of the whole configuration (the term "axial" may have been misleading here, and was therefore systematically replaced by "bilateral" or "mirror symmetry") on the perceived strength/perceptual salience of the figure-ground percept at the centre of the configuration. By definition, the configuration consists of a physically specified part, i.e. the inducers (the reviewer prefers to call them "pacmen", which is popular but, unfortunately, is not scientific as the term does not specify that they induce the perceptual phenomenon) AND a perceptually completed part, i.e. a perceived triangle in this case here. Therefore, symmetry of both the inducers and the triangle are necessary for bilateral symmetry of the whole configuration. Critical physical parameters are, as explained here above and in the text, held constant across conditions of symmetry/asymmetry, including the size of the triangle. As explained, the interpolation or contour support ratio was not varied in the triangular Kanizsa configurations AND the size of the induced triangular surface was also invariant (see Table 1). Therefore , what the reviewer calls "size of the illusory surface" or "2D compactness" of the configurations does not vary across symmetry conditions! There are no confounding factors, and the experiments are, indeed, well designed for what they aim for.
In the square version of the Kanizsa figure, for example, one cannot vary symmetry independently from shape and its resulting shape interpretation (in terms of perception) and the overall configural complexity. This is now explained in greater detail in the revised version of this manuscript (pages 11-13), with an additional Figure for illustrating this interesting problem. The triangular version of the Kanizsa figure was therefore singled out for this study and, indeed, effects of bilateral/mirror symmetry of the configuration on figure-ground strength/salience at the centre of the configuration are clear and non-ambivalent.
The results relative to interactions were added to the ANOVA tables.
All minor issues were, as requested, resolved in the revised manuscript. One table had, indeed, the wrong caption (fixed now), and the missing brackets were added to the equation. Many thanks for your time and valuable comments, which have been helpful.
Reviewer 3 Report
There is an extensive and good literature review of gestalt and grouping phenomena in the introduction.
Support ratio and physical-to-total contour ratio as defined by Shipley & Kellman is referred to in the reference but needs to be better defined when first introduced.
A clear set of hypotheses with regard to the three variables of interest (symmetry, orientation, and contrast polarity) needs to be stated at the end of the introduction. What is to be expected regarding base orientation or the different types of inducer polarities?
Thirty-two unique configurations were only presented once to each observer. Why weren't multiple presentations per observer obtained? If repeated observations of the same stimulus would bias judgements, I didn't pick this up.
Was there a large degree of variability between observers? Individual differences are not mentioned for the 10 participants.
In Figure 4 the standard error bar statistic isn't mentioned. Is this plus or minus one standard error of the mean? In Figure 5 it just says "error bar" in the caption.
I think more of an explanation is need for why there was no advantage for vertical symmetry in the discussion section. The author should go a little bit deeper into the literature here discussing the ecological significance of vertical and more recent studies that show no advantage for vertical.
The same is true for neural information processing models. Figure ground is hypothesized to be an early stage vision process, but so is symmetry detection, which has been found in some studies to be as fast as 50 ms. So these two probably occur in parallel. What is the significance of this for the results?
It would be nice to see in future studies a more quantitative manipulation of these or related variables so that the slope of the log-log estimation function could be measured and compared to other magnitude estimation studies like brightness.

Author Response
Thank you for your time and expertise. Your comments and suggestions have been fully taken into account in the revisions; please see here below my detailed replies to each issue raised:
“Support ratio and physical-to-total contour ratio as defined by Shipley & Kellman is referred to in the reference but needs to be better defined when first introduced.”
*Reply*: Please see the new text (on page 3 and later-on) of the revised manuscript. All changes are highlighted in green, for your attention. Necessary supplementary references were added.
“A clear set of hypotheses with regard to the three variables of interest (symmetry, orientation, and contrast polarity) needs to be stated at the end of the introduction.”
*Reply*: This request is also fully taken into account in the revision. Please see the new text (in green) at the end of the introduction
“Thirty-two unique configurations were only presented once to each observer. Why weren't multiple presentations per observer obtained? If repeated observations of the same stimulus would bias judgements, I didn't pick this up.”
*Reply*: There was no hypothesis regarding multiple presentations. Unique presentations generated a sufficient number of observations producing reliable effects.
“Was there a large degree of variability between observers? Individual differences are not mentioned for the 10 participants.”
*Reply*: It is now clearly stated (page 8 of the revised ms, in green) that the inter-individual consistency of the data was good (no “outlier” subjects). The individual data are made accessible to readers in the tables in the supplementary section.
“In Figure 4 the standard error bar statistic isn't mentioned. Is this plus or minus one standard error of the mean? In Figure 5 it just says "error bar" in the caption.”
*Reply*: This statistic is now made explicit in the figure legends (please see the new text added in green).
“I think more of an explanation is need for why there was no advantage for vertical symmetry in the discussion section. The author should go a little bit deeper into the literature here discussing the ecological significance of vertical and more recent studies that show no advantage for vertical.”
*Reply*: This suggestion was also fully taken into account. Please see the new text on pages 11 and 12, lines 368-389.
“The same is true for neural information processing models. Figure ground is hypothesized to be an early stage vision process, but so is symmetry detection, which has been found in some studies to be as fast as 50 ms, so, these two probably occur in parallel. What is the significance of this for the results?”
*Reply*: This is now discussed in detail on page 11, lines 351-367, of the revised manuscript
“It would be nice to see in future studies a more quantitative manipulation of these or related variables so that the slope of the log-log estimation function could be measured.”
*Reply*: Agreed! Future directions are now suggested in the new discussion section (in green).
Round 2
Reviewer 1 Report
I am happy with the revision. I have only some minor suggestions.
line 83 Typo "opimal limit"
line 93 as support ratio is defined here ("physically specified-to-total contour ratio, or support ratio") I suggest later in the paper only the short term "support ratio" can be used.
line 99 "the contrast polarity of these latter" -> " their contrast polarity"
line 224 "young individuals" Replace with either mean age or age range.
line 359 "in this study here" delete "here" (also in other places there are a few "here" that can be deleted)
Author Response
Thank you. All the minor amendments suggested were implemented into this second revision.
Reviewer 2 Report
In this revision of the manuscript, titled "Bilateral symmetry strengthens the perceptual salience of figure against ground", by Birgitta Dresp-Langley, the author responded all the comments raised by two reviewers. However, the main criticism (“It is impossible to discuss any role of symmetry based on only these two samples of the triangle shape.”) in my last review is not well addressed and I do not think this criticism can be addressed just by revising the text.
Allow me to be pedagogical. It is fundamental to systematically control stimuli in a psychophysical experiment. Note that a "shape" can be characterized by many different properties and they can introduce some unexpected effect on our perception. For example, Bahnsen (1928) showed effect of symmetry on figure-Ground organization but Kanizsa & Gerbino (1976) pointed out that the results of Bahnsen (1928) can be explained by an artifactual image property (convexity) of visual stimuli used in Bahnsen (1928). (To be sure, later studies have shown that both symmetry and convexity are effective cues for figure-ground organization.) Note that Bahnsen (1928) used only a few visual stimuli to test the effect of symmetry (as the experiments reported in this submitted manuscript). In many modern psychophysical studies, people use randomly generated stimuli with some constraints to minimize any artifactual effect attributed to these properties in a whole session statistically rather than at individual trials of the session. So, it is not justifiable to discuss any role of symmetry on visual perception based on only two triangle shapes.
Figure 6 and a part of the main text about Figure 6 (lines 398-414) are confusing. My best guess is that the author tries to discuss a relation between symmetry and rectangularity of an angle by showing that distorting a square also distorts its rectangular angles but it does not make sense. There are many quadrilaterals that are symmetric but do not have any rectangular angle (e.g. isosceles trapezoid, diamond). A symmetric isosceles trapezoid without any rectangular angles can be distorted to become an asymmetric (non-isosceles) trapezoid with two rectangular angles. We can make symmetric and asymmetric triangles with rectangular angles and symmetric and asymmetric triangles without rectangular angles (see theory of inscribed angle). So, I am not sure what kind of relation between symmetry and rectangularity the author tries to discuss here.
Author Response
I beg to differ from this reviewer's extreme standpoint on number of shapes. The number or quantity of stimuli used in an experiment is irrelevant, only the novelty of the data and the results, with respect to what the experiment was designed to show, matter. The text of my paper explains in great detail that the stimuli for the two experiments were very carefully chosen. They are the most classic examples of figure-ground from 2D occlusion cues in the Gestalt literature. The surfaces here, as explained in great detail in the manuscript (see also the new text added in green, which makes this point even clearer now), result from local occlusion cues exploited by brain mechanisms driving a functionally identified spatial interpolation process across edge fragments for surface completion (von der Heydt, 2016; Erlikhman & Kellman, 2015; 2016).
The results in this paper here clealry show, and for the first time as far as I know, a significant contribution of symmetry to this specific process of perceptual organization. What is reported here therefore is meaningful and complete within the scope of the manuscript. The data from the two experiments are clear and consistent. The role of convexity was not studied here, nor is there any reason why it should have been. As this reviewer points out, the question of convexity was addressed elsewhere by others and is not the topic of this study here.
It is quite unclear why this reviewer makes unnecessary guesses about Figure 6. In the manuscript text and in the Figure legend it is clearly stated that Figure 6 illustrates some very specific configurational constraints in the Kanizsa square. These explain why it was not chosen for the experiments here - assumptions about a relationship between symmetry and rectangularity are neither made here, nor are they implied.